# Comparative Analysis of Mitochondrial Genome Features among Four *Clonostachys* Species and Insight into Their Systematic Positions in the Order Hypocreales

**DOI:** 10.3390/ijms22115530

**Published:** 2021-05-24

**Authors:** Zhiyuan Zhao, Kongfu Zhu, Dexiang Tang, Yuanbing Wang, Yao Wang, Guodong Zhang, Yupeng Geng, Hong Yu

**Affiliations:** 1College of Ecology and Environmental Sciences, Yunnan University, Kunming 650091, China; zhiyuan@mail.ynu.edu.cn; 2The International Joint Research Center for Sustainable Utilization of Cordyceps Bioresources in China and Southeast Asia, Yunnan University, Kunming 650091, China; zwkongfu@gmail.com (K.Z.); tang2255@mail.ynu.edu.cn (D.T.); wangyb001@126.com (Y.W.); wangyao1@aliyun.com (Y.W.); zhangguodong_zyx@163.com (G.Z.)

**Keywords:** *Clonostachys*, mitochondrial genome, protein coding genes, repeat sequence, phylogenetic analysis

## Abstract

The mycoparasite fungi of *Clonostachys* have contributed to the biological control of plant fungal disease and nematodes. The *Clonostachys* fungi strains were isolated from *Ophiocordyceps highlandensis*, *Ophiocordyceps*
*nigrolla* and soil, which identified as *Clonostachys*
*compactiuscula*, *Clonostachys*
*rogersoniana*, *Clonostachys*
*solani* and *Clonostachys* sp. To explore the evolutionary relationship between the mentioned species, the mitochondrial genomes of four *Clonostachys* species were sequenced and assembled. The four mitogenomes consisted of complete circular DNA molecules, with the total sizes ranging from 27,410 bp to 42,075 bp. The GC contents, GC skews and AT skews of the mitogenomes varied considerably. Mitogenomic synteny analysis indicated that these mitogenomes underwent gene rearrangements. Among the 15 protein-coding genes within the mitogenomes, the nad4L gene exhibited the least genetic distance, demonstrating a high degree of conservation. The selection pressure analysis of these 15 PCGs were all below 1, indicating that PCGs were subject to purifying selection. Based on protein-coding gene calculation of the significantly supported topologies, the four *Clonostachys* species were divided into a group in the phylogenetic tree. The results supplemented the database of mitogenomes in Hypocreales order, which might be a useful research tool to conduct a phylogenetic analysis of *Clonostachys*. Additionally, the suitable molecular marker was significant to study phylogenetic relationships in the Bionectriaceae family.

## 1. Introduction

*Clonostachys* (Bionectriaceae, Hypocreales) refers to an asexual morph-typified genus formed by the type species *C. araucaria* in 1939, which is now recognized as a synonym of *C. rosea* [1]. Most *Clonostachys* species inhabit soil, fungi and dying trees or rotting leaves in tropical and temperate regions [2]; minorities are also parasitic to myxomycetes, nematodes, ticks, mollusc and spiders [3,4]. The genus has been extensively studied, with 92 names recorded (http://www.indexfungorum.org, accessed on 10 December 2020) [3,5,6,7]. Some strains of *Clonostachys* have been commercially developed for the biological control of plant pathogens. *C. rosea* is a generally well-known species of the genus and can potentially contribute to the biological control of plant fungal disease and nematodes [8].

Most members of *Clonostachys* genus are transferred from another genus, such as *Bionectria*, *Nectria* and *Gliocladium* genus. However, this genus taxonomic position, compared with another family in the Hypocreales order, is often debated [9]. Since Schroers et al. (2001), based on Tub and ITS rDNA sites, arranged *Bionectria* and *Clonostachys*, 43 species were reported in this genus [3]. A growing number of new species were reported or revised using different molecular markers to modify the *Clonostachys* genus [1,6]. James et al. (2006) reported Bionectriaceae as a sister taxa with the family Nectriaceae, with Hypocreaceae in a basal position, whereas Sung et al. (2007) reported the Bionectriaceae Foundation to both Nectriaceae and Hypocreaceae. Although a lot of molecular markers have been used, the database of TUB, ITS and nrLSU sites still proved efficient in the traditional identification of *Clonostachys* species [10]. A lack of data and heterogeneity are disadvantages in the phylogenic analysis research of *Clonostachys* species. Accordingly, it is imperative to acquire more useful data and use advanced techniques to illustrate the taxonomic status and characteristics of this genus.

The mitochondrion, a circle organelle exists in almost eukaryotes species cell, which usually provides a source of chemical energy through oxidative phosphorylation [11]. The mitochondrion exhibits many advantages, such as their evolutionary rate being faster than nuclear DNA, the characteristics of conservation gene functions, their high copy number and AT content [12]. Moreover, single genes can make it difficult to analyze the interpretation of fungus relationship, as they are impacted by their different evolution rates [13]. For this reason, the complete mitogenome sequence is not only considered an advanced tool for the population genetics and phylogenetic research, but also for deeply understanding the systematic genetics of fungal. However, *Clonostachys* rosea has been the only reported mitogenome of *Clonostachys* to date [14]. The lack of data is a limitation to phylogenic analysis research of *Clonostachys* species. Therefore, it is necessary to complement the mitogenome of the *Clonostachys* species.

In this study, four *Clonostachys* species were accurately identified, i.e., *Clonostachys compactiuscula*, *Clonostachys rogersoniana*, *Clonostachys solani* and *Clonostachys* sp. This was used to characterize morphologies under a microscopy and to conduct molecular analyses. The *Clonostachys* sp. refers to a potential new species of *Clonostachys* genus. Further, the four complete mitogenomes were sequenced, assembled and annotated. In order to identify the phylogenetic relationship and sequence characteristics of *Clonostachys* genus, these four mitogenomes’ gene contents, gene orders and codon usage were analyzed comparatively. The structural differences in the mitochondrial genomes in this genus were analyzed. The results achieved in this study not only complement new molecular information about the Bionectriaceae family in the Hypocreales order, but also provide an understanding of the characteristics and phylogenetic relationships of this class.

## 2. Results

### 2.1. Identification of Four Clonostachys Species

To ensure species accuracy, microscopic morphological characteristics and molecular analyses were first used to identify the abovementioned species. Micromorphological characteristics with asexual morphs involving *C. compactiuscula*, *C. rogersoniana* and *C. solani* were compared with the original description [3,15] (Figure 1).

Phylogenetic analyses based on the combined 3-gene (*nrLSU* + *Tub* + *ITS*) database, consisting of 66 fungal sequences, as well as ML analyses and BI analysis, were conducted to verify the presence and positions of the four *Clonostachys* species. The phylogenetic tree demonstrating that *C. rogersoniana*, *C. compactiuscula* and *C. solani* join the reference sequence come from NCBI and exhibited high support rates. In addition, the *Clonostachys* sp. was an unknown species based on microscopic morphological and phylogenetic analyses (Figure 2).

### 2.2. Characteristics of the Mitogenome of Four Clonostachys Species

The complete mitogenome of four *Clonostachys* species are typical circular molecule with a total length ranging from 27,410 to 42,075 bp (Figure 3). Mitogenome sizes varied extensively among the four mitogenomes, with the largest mitogenome from *C. rogersoniana* being 1.54 times larger than the smallest mitogenome from *Clonostachys* sp. The GC content in the four mitogenomes ranged from 25.89% to 28.54%, among which the GC content in the genome of *C. solani* was the highest. Except for *C. rogersoniana*, the other three mitogenomes were negatively AT-skewed. However, all species suggested that mitogenomes were positive GC skew, with a skew value of 0.1265 to 0.1448. Appendix A lists the proportion of genomic regions, tRNA, rRNA genes, intronic regions, and intergenic regions.

The numbers of PCGs were different in four species: *C. compactiuscula* included 23 PCGs, 35 in *C. rogersoniana*, 22 in *C. solani*, 16 in *Clonostachys* sp. The total species also contained the 15 core PCGs, except for rps3 which participate in transcriptional regulation, and another fourteen genes, i.e., atp6, atp8, atp9, cob, cox1, cox2, cox3, nad1, nad2, nad3, nad4, nad4L, nad5, and nad6, were covered in energy metabolism. Moreover, several (Open Reading Frames (ORFs) were identified in the four mitogenomes, in amounts ranging from 1 to 20 in each genome. Except for *orf178* of *C. solani*, located on the reverse strand, all the genes of another three mitogenomes were located on the direct strand.

The four *Clonostachys* mitogenomes contained overlapping nucleotides sequences ranging from 1 to 3 bp, and each species contained the same gene overlapping region between the adjacent genes *nad4L*- *nad5* (1 bp). The *C. compactiuscula* mitogenomes identified that largest overlapping nucleotide was 2056 bp, located at neighboring genes rnl and rps3. We identified intergenic nucleotides of 8383 bp, 3807 bp, 8162 bp and 6096 bp from mitogenomes of *C. compactiuscula*, *C. rogersoniana*, *C. solani* and *Clonostachys* sp. The length of intergenic segments ranged from 1 to 1358 bp. The longest intergenic segments were situated in *orf394* and trnR-TCT of *C. rogersoniana*.

Each of the four *Clonostachys* mitogenomes involved two rRNA genes: the small subunit ribosomal RNA gene (*rns*) and the large subunit ribosomal RNA gene (*rnl*). No significant difference was identified in the number of rRNA genes in the four species, and the total length of rRNA genes in the four *Clonostachys* mitogenomes was altered from 6163 bp to 6400 bp. Besides *C. rogersoniana*, where 26 tRNA genes were identified in the mitogenomes, 25 tRNA genes were identified in the other three mitogenomes, encoding 20 standard amino acids. Each of the mitogenomes contained three tRNA genes with the identical anticodons for methionine (trnM-CAT), and two tRNAs with different anticodons for arginine (trnR-ACG and trnR-TCT) and serine (trnS-GCT and trnS-TGA). Remarkably, *C. rogersoniana* mitogenome included four different anticodons for leucine (trnL-TAA and trnL-TAG, trnL-AAG, trnL-CAG). In another three mitogenomes, *C. compactiuscula*, *C. solani* and *Clonostachys* sp. included two for leucine (trnL-TAA and trnL-TAG). The length of tRNA genes in the four mitogenomes ranged from 71 bp to 86 bp (Appendix A).

The maximum proportions of the four *Clonostachys* mitogenomes are protein-coding regions, taking up 53.9–69.4% of the whole mitogenomes. The RNA and intergenic segments accounted for 20.4–29.5%, 6.8–20.2%, respectively. The *C. compactiuscula* mitogenome consisted of four intronic regions and *C. rogersoniana*, *C. solani* covered two intronic regions, accounting for 6.2–9.4% (Figure 4).

As was revealed from the codon usage analysis, the most frequently used codons in the four mitogenomes included TTT (for phenylalanine; Phe), TTA (for leucine; Leu), ATT (for isoleucine; Ile), TAT (for tyrosine; Tyr), AAT (for asparagine; Asn), AAA (for lycine; Lys), as well as ATA (for isoleucine; Ile) (Figure 5 and Appendix A). The high AT content of the four mitogenomes (mean: 73.15%) was attributed to the high frequency of AT content in these codons.

### 2.3. Repetitive Element Analysis

Fifteen repeat sequences were identified in *C.*
*solani*, 13 in *C.*
*compactiuscula* and 2 in *Clonostachys* sp., except for *C**. rogersoniana* mitogenome; the test used self-comparisons of mitogenomes BLASTn searches (Appendix A). The length of these repeats ranged from 33 bp to 142 bp, with pair-wise nucleotide similarities ranging from 83% to 100%. The widest repeat region of the *Clonostachys* sp.’ mitogenome was 142 bp, 116 bp in *C. compactiuscula* and 123 bp in *C. solani*, comprised of the largest scale of repeat regions, taking up 3.06% of the whole mitogenome, followed by *C. compactiuscula* (2.18%), and *Clonostachys* sp. (2.11%).

A total of three tandem repeats were detected in the *C. rogersoniana*, 15 in *C. solani*, 6 in *C. compactiuscula* and 8 in *Clonostachys* sp. (Appendix A). The longest repeat sequence reached 83 bp in *C. rogersoniana*, 85 bp in *C. solani*, 92 bp in *C. compactiuscula* and 143 bp in *Clonostachys* sp. Most of the tandem repeat sequences were duplicated once or twice in each of the respective mitogenome. A total of 15 forward, 21 reverse repeats and 10 palindromic were measured with REPuter in the *Clonostachys* sp.’s mitogenome (Appendix A). Repeat sequences accounted for 0.3%, 2.2%, 0.88% and 1.76% of the mitogenome from *C. rogersoniana*, *C. solani*, *C. compactiuscula* and *Clonostachys* sp., respectively. 

### 2.4. Variation and Genetic Distance of Core Genes

In the four *Clonostachys* species, eleven variable lengths were found across the 15 core PCGs. The *atp8*, *nad3*, *nad4L* and *cox3* genes are the same length in the four *Clonostachys* mitogenome. Furthermore, there were significant differences in GC content and the *atp9* gene had the highest GC content of the 15 core PCGs. Most of the PCGs involved negative AT skews, with the exception of *rps3* of all species, *atp6* of the *C.*
*rogersoniana*, nad4L of *C. rogersoniana* and *Clonostachys* sp., which exhibited a positive AT skew. However, *nad4L of C. compactiuscula* does not generate an AT skew, as the impacts from the equivalent base A and T content were equivalent. The GC skew in 15 core PCGs varied significantly in the four *Clonostachys* mitogenomes. The *cox3* gene in *C. rogersoniana*, atp8 gene in *Clonostachys* sp., *C.*
*compactiuscula and C. rogersoniana* displayed negative GC skews, while other genes showed a positive GC skew among the mitogenomes (Figure 6).

Of the 15 core PCGs, nad6 displayed the closest K2P genetic distance among the four mitogenomes, followed by *rps3* and *cox3*, suggesting that these genes evolved faster than other genes. The genetic distance of *nad4L* among the mitogenomes was the minimum of the 15 core PCGs, indicating that *nad4L* is highly conserved across the mitogenomes of the four *Clonostachys* species (Figure 7). The highest nonsynonymous substitution rate (Ka) was rps3, and the *atp6* was the lowest of the 15 core PCGs. The nonsynonymous substitution rate (Ks) was highest in nad1, and *nad4L* had the lowest rate of the 15 core PCGs. The Ka/Ks values indicating that all PCGs were subjected to purifying selection were due to the Ka/Ks values of 15 PCGs, which were <1.

### 2.5. Gene Arrangement Analysis

Gene order and rearrangement in the mitogenome had great significance for biological evolution. The relative positions of two rRNA and fifteen PCGs were highly conserved across the four *Clonostachys* mitogenomes, showing a close phylogenetic relationship among the mentioned species. Additionally, the order of 24 tRNAs shared by the four *Clonostachys* species were identical across the four mitogenomes. However, only *C. rogersoniana* exhibited two additional tRNAL, thereby causing the difference in the gene orders among the four mitogenomes.

Gene Synteny analysis results revealed that six homologous regions (A-F) were divided into four *Clonostachys* mitogenomes. The relative positions of regions were highly conserved, and four of the six were detected in these species mitogenomes, while only *C. solanni* contained whole regions compared to *C. rogersoniana*, *C. compactiuscula and*
*Clonostachys* sp. However, the size of these homologous regions varied considerably across the four *Clonostachys* mitogenomes. The widest homologous region was region A in *C. compactiuscula and C. solanni,* making the mitogenome sequence longer than other mitogenomes. The *Clonostachys* sp. mitogenome lacked two regions for E and F, which resulted in a shorter mitogenome sequence than other three mitogenomes (Figure 8).

### 2.6. Phylogenetic Analysis

The phylogenetic analysis employed the nucleotide sequences of 14 common core genes from these four *Clonostachys* mitogenomes, as combined in 52 species that represent six families in the Hypocreales order, i.e., Ophiocordycipitaceae, Clavicipitaceae, Hypocreaceae, Cordycipitaceae, Bionectriaceae and Nectriaceae. For both the ML and BI trees, the 52 Hypocreales species could be divided into six major clades, which coincidently classified with different clades and exhibited high confidence (BI value of 1 and bootstrap value of 100%). Among the species in Hypocreales order, the Nectriaceae family was located at the foundational position of the integral phylogenomic tree. The clade united at the families of Ophiocordycipitaceae and Clavicipitaceae, and formed a sister cluster with the families of Hypocreaceae and Cordycipitaceae. Furthermore, the *Clonostachys* species comprised the Bionectriaceae clade and was further subdivided into groups, recovering as (*Clonostachys* sp. + (*C.compactiuscula+* (*C. rosea* + *C. solanni*))) (Figure 9). The *C. rosea* was identified as a sister species to *C. solanni,* complying with the 3-gene (*nrLSU* + *Tub* + *ITS*) database and existing studies.

## 3. Discussion

To carry out a good systematic study of this genus, a new potential species was found, and the mitogenomes of four species in *Clonostachys* genus were published, which provided more insights into mitogenomes in this genus and supplemented the molecular database in such a classification group.

As compared with animal and plant mitogenomes, fungal mitogenomes are the largest and most variable length in eukaryotic mitogenomes [15,16]. However, fewer than 500 fungal species mitogenomes were reported (https://www.ncbi.nlm.nih.gov/genome/browse#!/organelles/, accessed on 10 December 2020, within the scope of 1140 bp [17] to 258,879 bp [18]. The size difference between the two species is 236 times, and the largest and smallest mitogenomes were *Rhizoctonia*
*solani* and *Hyaloraphidium curvatum*, respectively [17,18]. Likewise, variability in mitogenome size was regularly identified in the Hypocreales order. The mitogenomic length of *Ophiocordyceps sinensis* was the largest mitogenomic (157,539 bp) in the Hypocreales order, containing the most introns [19]. Previous studies suggested that the variations in mitogenomic length should primarily be correlated with the presence of new genes through horizontal transfer, the number and size of introns, and the frequency of duplicated repeats [20,21,22,23]. In the present study, the lengths of mitogenomes in *Clonostachys* were varied, ranging from 27,410 bp to 42,075 bp. The considerable variation in mitogenome sizes provided a new insight into mitogenome differences at the genus level [24]. However, *C. compactiuscula*, *C. rogersoniana* and *C. solani* mitogenomics contained two introns, except for *Clonostachys* sp. In the present paper, *Clonostachys* sp. mitogenomic was the shortest among all *Clonostachys* species’ mitogenomics. Additionally, the mitogenome’s size was not always positively related to the number of introns. For example, the length of *C. rogersoniana*’s mitogenomic was the biggest among the four species due to its adding potential new genes. The GC content of mitogenomics is considered to be affected by the mutation bias, selection and biases of reconstitution-related DNA repair, which were clearly differentiated [25,26,27]. Here, the GC content of four mitogenomes was observed to be highly variable at the genus level (25.89–28.54%). However, without mutation or selection bias [28], every base in the complementary DNA strand exists at an approximately identical frequency on the basis of the second parity rule. Nonetheless, base biases are discovered in the eukaryotes’ mitogenomes as well [24]. Subsequently, the three AT skews of the four *Clonostachys* species were negative (A less than T), while the other four were positive (T less than A), suggesting that these four species might suffer different selection pressures.

Since the mitogenome of eukaryotes obtained from ancestral endosymbiotic bacteria, mitogenomes have been considerably differentiated with original nuclear genomes [29,30], in which the number of mitochondrial genes was transferred to the eukaryotic nuclear genome in the evolution [31,32,33]. This phenomenon demonstrates the many advantages conferred [34]. However, some genes were obtained and transferred from nuclear genomes in the mitochondria, which might promote localized control, for the purposes of redox regulation, over the mitochondrial machinery [35,36]. The 15 core PCGs, *atp6*, *atp8*, *atp9*, *cob*, *cox1*, *cox2*, *cox3*, *rps3*, *nad1*, *nad2*, *nad3*, *nad4*, *nad4L*, *nad5* and *nad6*, were observed in most eukaryotic mitogenomes. The fifteen core genes differed considerably in length, GC content, and evolutionary rates among the mitogenomes of the *Clonostachys* species. Moreover, the AT and GC skews were varied among species in these core genes, which facilitated the observed genetic differentiation among the mentioned species. Moreover, some independent and unknown functions ORFs were detected in *Clonostachys* mitogenomes and these genes’ sizes and position varied among the mitogenomes of *Clonostachys*, indicating that there were still many undiscovered proteins in *Clonostachys* mitogenomes. Consequently, these ORFs were important factors that contributes to mitogenome differentiation, such as gene rearrangement among *Clonostachys* species. These results were consistent with previous studies [37].

Repetitive elements often change the mitochondrial structure in the fungal mitogenome [38]. The changes in gene order and generation of new genes are influenced by these phenomena [29]. Furthermore, the loss and increase in genes could be accompanied by the accumulation of repeated elements. In this study, repeated sequences were identified in three *Clonostachys* mitogenomes, except for *C. rogersoniana*. The length of these repeats was ranged from 33 bp to 146 bp. The repeated sequences found in the three mitogenomes might cause variations in gene structure and content, or with contribute to species’ differentiation.

An increasing number of molecular markers were adopted for phological analysis, fueled by the advances in sequencing technology. However, phylogenetic studies are still at the adverse state due to species heterogeneity and lack of data. Therefore, finding effective molecular markers is of high importance for the *Clonostachys* genus. Mitogenomes have many advantages, such as their rapid evolution, few genes, simple structure, maternal inheritance and single copy [39], which are extensively exploited in research into population genetics, comparison and evolution genomics, phylogenetics and conservation biology. In the present study, the Hypocreales phylogenetic tree was built based on 14 common PCGs and was consistent with traditional morphological classifications. However, in previous studies, the single genes of *cox1*, *cox2* and *c**ob* were often used in some species phylogenetic research, especially in animals [40,41]. Furthermore, *cox1* can be considered as a potential molecular marker for the phylogenetic analysis of fungal species. The results achieved in this study provide a useful tool for phylogenetic analysis of *Clonostachys* genus. The suitable molecular marker is significant to study phylogenetic relationships in the Hypocreales order.

## 4. Materials and Methods

### 4.1. Sample Collection, Identification and DNA Extraction

Four species, *C. solani* and *Clonostachys* sp. were isolated from *Ophiocordyceps highlandensis*, *C. compactiuscula* was isolated from *Ophiocordyceps*
*nigrolla* and *C. rogersoniana* was isolated from soil in Kunming, China. The fungi were transferred onto plates of potato dextrose agar (PDA; potato 200 g/L, dextrose 20 g/L, agar 20 g/L) and subsequently cultured at 25 °C. The species were tested according to their microscopic morphological characteristics and through molecular analyses [42,43,44] and microscopic morphological observation methods, as described in Wang et al. (2020). Molecular analyses used a 3-gene (nrLSU + Tub + ITS) database, which consisted of 64 sequences of *Clonostachys* genus, and the outgroup comprised *Stanjemonium ochroroseum* and *Stanjemonium grisellum*. Four specimens were deposited as vouchers in the Yunnan Fungi Culture Collection (YFCC) at the Institute of Herb Biotic Resources of Yunnan University with the registration numbers of YFCC 8593, YFCC 8591, YFCC 8592 and YFCC 895, respectively. Genomic DNA extraction was conducted with a fungal DNA kit (#D3390-00, Omega Bio-Tek, Norcross, GA, USA), complying with the instructions of the manufacturer. The integrity of the genomic DNA was measured by 1% agarose gel electrophoresis, and the concentration was measured using NanoDrop (Wilmington, DE, USA).

### 4.2. Sequencing, Assembly, and Annotation of Mitogenomes

The Illumina TruSeq library was generated using the IlluminaTruseq™ DNA Sample PreparationKit (BGI, Shenzhen, China), following the manufacturer’s instructions. The sequencing of four genomes was conducted on Illumina HiSeq 4000 Platform for a PE2 × 150 bp sequencing and the reads were controlled with an average quality lower than Q20. After removing unpaired, short and low-quality reads, clean reads were used. Next, the reads of mitogenomes were collected and expended of mitogenomes using by Getorganelle 1.6.4 and the clean data were assembly mitogenomes with the SPAdes 3.13.0 software package [45,46,47,48,49]. The four complete *Clonostachys* mitogenomes were initially annotated by MFannot [50] (https://megasun.bch.umontreal.ca/RNAweasel/, accessed on 10 December 2020) and MITOS [51] (http://mitos2.bioinf.uni-leipzig.de/index.py, accessed on 10 December 2020) under the genetic code (genetic code Appendix A) in both programs. After this, using the NCBI Open Reading Frame Finder (https://www.ncbi.nlm.nih.gov/orf, accessed on 10 December 2020 [52], tRNAscan-SE 2.0.5 Search Server (http://lowelab.ucsc.edu/tRNAscan-SE/, accessed on 10 December 2020), four mitogenomes were inspected and annotated integrally. The PCGs’ intron-exon borders were checked by exonerate 2.2.0 software package [53] and graphical maps of four complete mitogenomes were generated by OGDraw (https://omictools.com/ogdraw-tool, accessed on 10 December 2020) [54].

### 4.3. Sequence Analyses of Mitogenomes

The base composition analysis of the four mitogenomes was conducted using DNASTAR Lasergene 7.1. The nucleotide length, composition, and codon usage of the complete mitogenomes of *C. compactiuscula*, *C. rogersoniana*, *C. solani* and *Clonostachys* sp. were computed using MEGA 6 [55]. The difference in base composition and strand asymmetry was determined by GC and AT-skews. This was measured based on the formulas: AT skew = [A − T]/[A + T]; GC skew = [G − C]/[G + C] [56]. The codon usage frequency was calculated using the Codonw 1.4.4 (http://codonw.sourceforge.net/, accessed on 10 December 2020) [57]. The overall mean genetic distances among the 15 core PCGs of the *Clonostachys* species were determined according to Kimura-2-parameter (K2P) substitution model in MEGA6.0 [55]. The nonsynonymous (Ka) and synonymous substitution rates (Ks) among the 15 core PCGs of the mitogenomes were calculated using KaKs_Calculator2.0 [58], and the code type was found using the Mold Mitochondrial Code. Lastly, genomic synteny of the four mitogenomes was analyzed with Mauve v2.4.0 [59].

### 4.4. Repetitive Elements Analysis

To determine whether these mitogenomes underwent intra-genomic duplication or interspersed repeats of large fragments, we used an E-value of <10^−10^ of BLASTn searches of each mitogenome [60]. Tandem repeats in the mitogenomes were identified with the online website tool (http://tandem.bu.edu/trf/trf.advanced.submit.html, accessed on 10 December 2020) [61], based on default parameters. Additionally, forward (direct), reverse, complemented, and palindromic (reverse complemented) repeats were identified with the REPuter (https://bibiserv.cebitec.uni-bielefeld.de, accessed on 10 December 2020) [62] software with default settings and E-values of <10^−5^.

### 4.5. Phylogenetic Analysis

In order to determine the phylogentic location of four *Clonostachys* species, phylogenetic analyses were conducted the homologous of 14 protein-coding genes (PCGs) in the complete mitogenomes. In total, this consisted of 52 sequences downloaded from the GenBank, including among the published sequences of Hypocreales order. Furthermore, *Podospora anserina* (X55026) was classified as the outgroup [63]. The amino acids and nucleic acid sequence were aligned using Clustal X version 2.0 [64] with the default parameters, respectively. Phylogenetic relationships were conducted for the Bayesian inference (BI) and maximum likelihood (ML) analyses with MrBayes v3.2.6 [65] and RAxMEL version 8.1.12 [66,67]. The MtREV + I + G + F model was selected in RAxML version 8.1.12 for the amino acid and the BI analysis selected a proportion of invariant sites of 0.01 and a gamma distribution shape parameter of 0.031 under the MrBayes v3.2.6. The first 25% of the trees were discarded as burn-in, and four chains of simultaneous were operated for 10,000,000 generations. The branch support of the dataset was determined by 1000 ultrafast bootstrap replicates.

### 4.6. Data Availability

The four newly sequenced mitogenomes were submitted to the GenBank database under the accession numbers of *C. compactiuscula* (MW030498), *C. rogersoniana* (MW030499), *C. solani* (MW0304665), and *Clonostachys* sp. (MW0304666). The GenBank accession numbers of mitogenomes sequences are provided in Appendix A. The ITS, Tub and nrLSU’s accession numbers of *C. rogersoniana*, *C. compactiuscula*, *Clonostachys* sp. and *C. solani* are referred to in Appendix A.

## Figures and Tables

**Figure 1 ijms-22-05530-f001:**
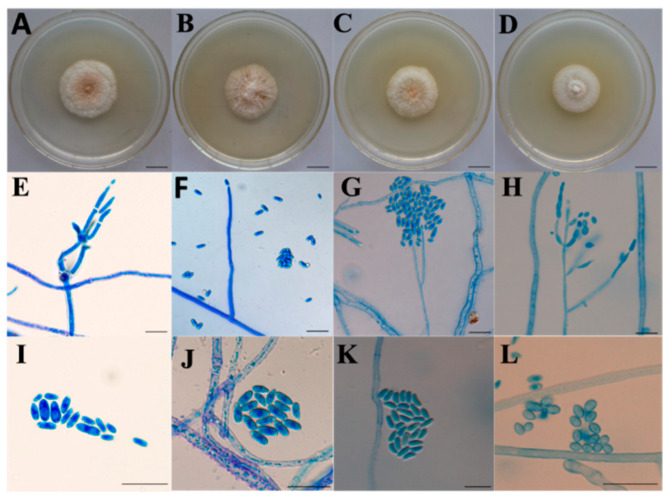
Culture and micromorphological characteristics with asexual morphs of four *Clonostachys* species in this study. (**A**–**D**): Culture of *C. compactiuscula*, *C. rogersoniana*, *C. solani* and *Clonostachys* sp.; (**E**–**H**): Perithecia of *C. compactiuscula*, *C. rogersoniana*, *C. solani* and *Clonostachys* sp.; (**I**–**L**): Conidia of *C. compactiuscula*, *C. rogerso**niana*, *C. solani* and *Clonostachys* sp. Scale bars (**A**–**C**) = 1.5 cm; (**D**) = 1 cm; (**E**–**H**) = 10 μm; (**I**) = 2.5 μm; (**J**–**L**) = 10 μm.

**Figure 2 ijms-22-05530-f002:**
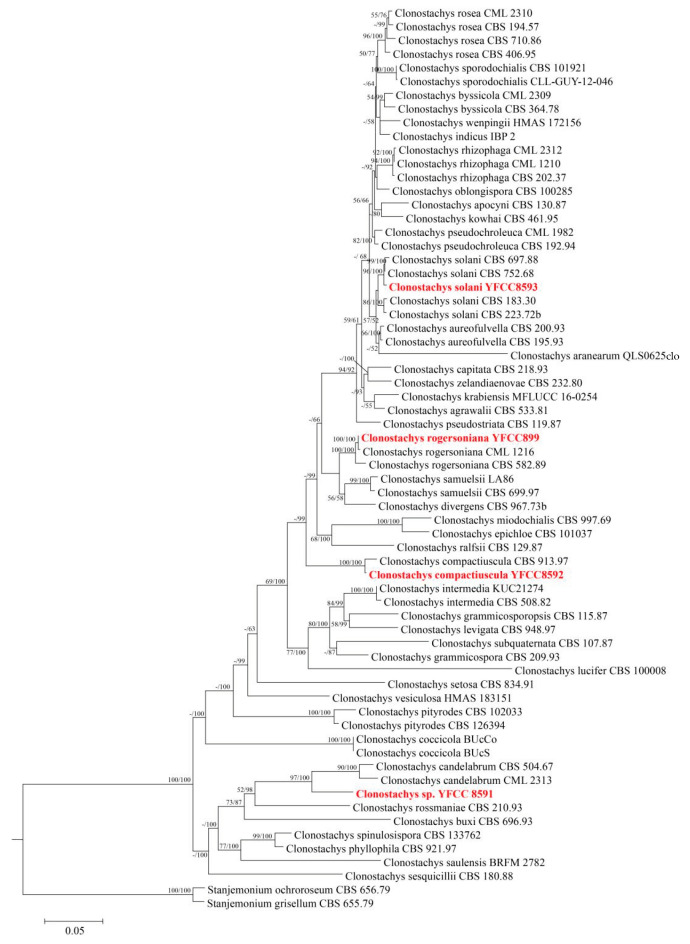
Phylogenetic tree of *Clonostachys* genus by the analysis of three genes (*nrLSU* + *Tub* + *ITS)* dataset.

**Figure 3 ijms-22-05530-f003:**
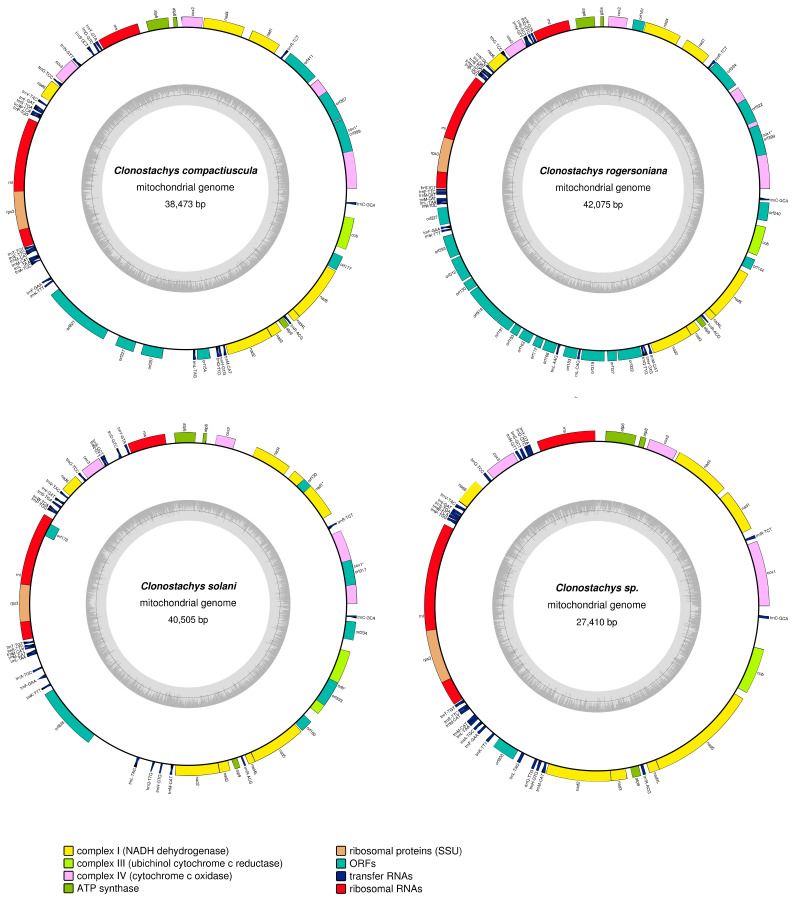
Circular maps of four newly sequenced mitogenomes from different *Clonostachys* species. (Genes are represented by different colored blocks, as presented in the legend below the maps. Colored blocks outside of each ring indicate that the genes were on the direct strand, while colored blocks in the rings indicate that the genes were located on the reverse strand.).

**Figure 4 ijms-22-05530-f004:**
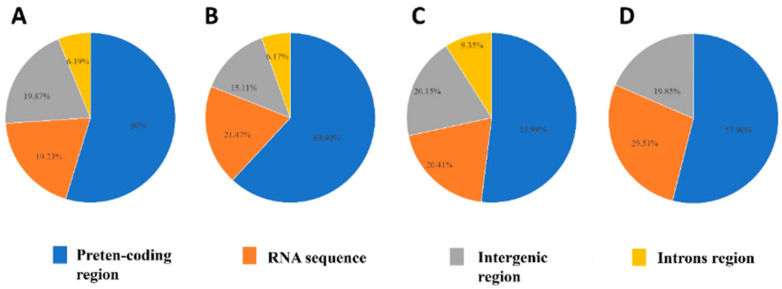
The proportion of the mitogenomes comprised by protein-coding, intronic, intergenic, and RNA genes (tRNA and rRNA) for four *Clonostachys* species. ((**A**–**D**): *C. compactiuscula*, *C. rogersoniana*, *C. solani*, *Clonostachys* sp.).

**Figure 5 ijms-22-05530-f005:**
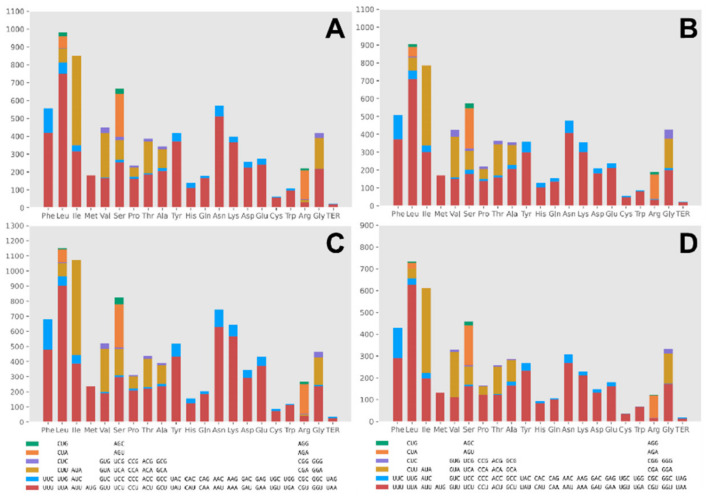
Codon usage analysis of the mitochondrial genomes from four *Clonostachys* species. ((**A**–**D**): *C. compactiuscula*, *C. rogersoniana*, *C. solani*, *Clonostachys* sp.).

**Figure 6 ijms-22-05530-f006:**
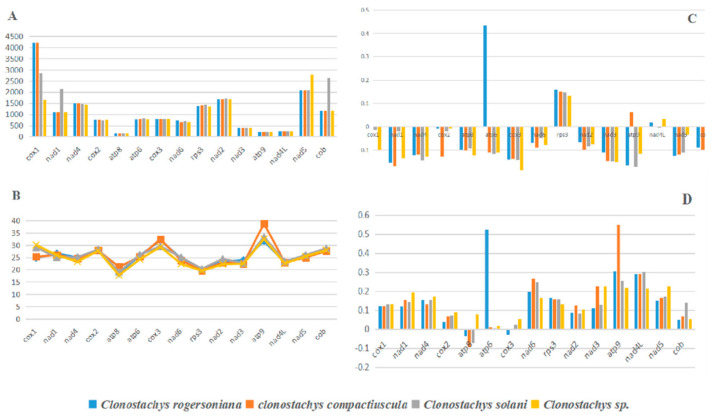
Variation in length and base composition of each of the 15 core protein-coding genes (PCGs) among four *Clonostachys* mitochondrial genomes. ((**A**), PCG length variation; (**B**), GC content across PCGs; (**C**), GC skew; (**D**), AT skew).

**Figure 7 ijms-22-05530-f007:**
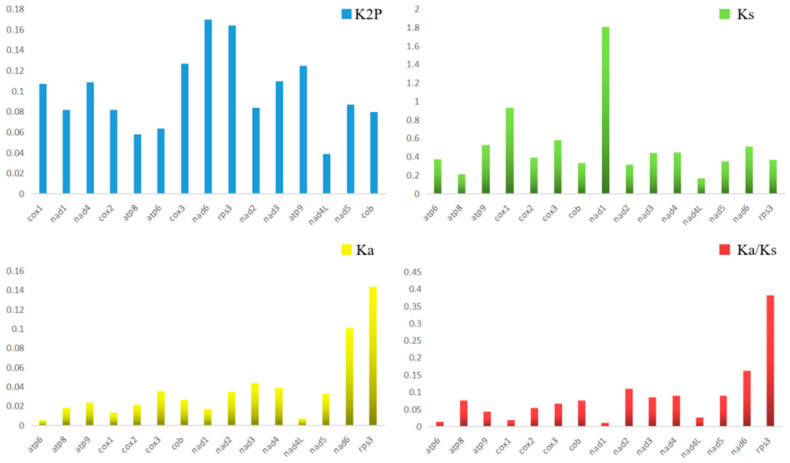
Genetic analysis of the 15 core protein-coding genes among four *Clonostachys* mitogenomes. (K2P, the Kimura-2-parameter distance; Ka, the number of nonsynonymous substitutions per nonsynonymous site; Ks, the number of synonymous substitutions per synonymous site.)

**Figure 8 ijms-22-05530-f008:**
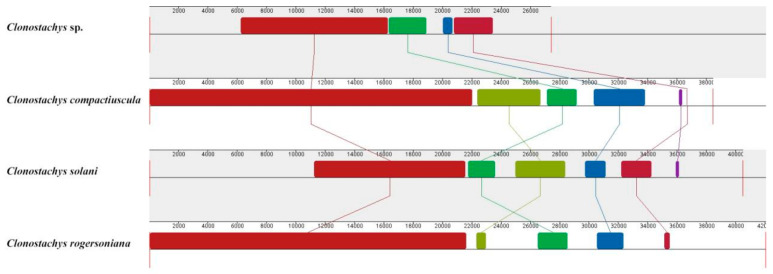
Mitogenome synteny among four *Clonostachys* species. (Six homologous regions were identified among the four mitogenomes, while the sizes and relative positions of the homologous fragments varied across the mitogenomes).

**Figure 9 ijms-22-05530-f009:**
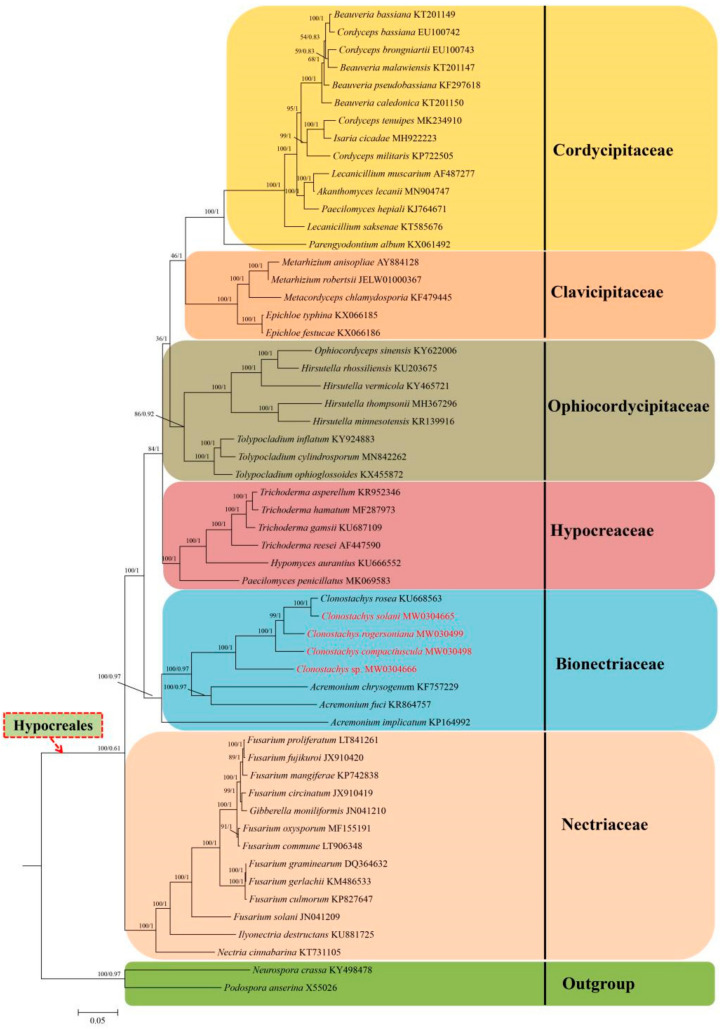
Molecular phylogenies of 54 species based on Bayesian inference and Maximum likelihood analysis of 14 protein-coding genes (PCGs).

## Data Availability

Data is contained within the article or Appendix A.

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
