# Peer review of "Comparative Analysis of Mitochondrial Genome Features among Four Clonostachys Species and Insight into Their Systematic Positions in the Order Hypocreales"

_ijms, 2021, doi:10.3390/ijms22115530_

Round 1

Reviewer 1 Report

The paper describes a study on sequencing and analysis of the mitochondrial genomes of four Clonostachys species. Unfortunately, the manuscript is poorly prepared and contains a lot of errors and mistakes in terms of grammar, style and overall language quality. This often makes the sentence hard to understand and following the story is almost impossible. Major complaints and suggestions are given below.

There are numerous errors in the Abstract. This will not attract the attention of the potential readers to the article.

Introduction is compact and highlights the subject of the study. Again, language quality needs to be improved.

Results: Figure 3 is not visible even under 400% magnification. Consider changing the way of presenting these results. Improve the descriptions of Figures 4, 5 and 6 (preten-coding region, B-D panel description in Fig.5, visibility, A-D panel description, Y axis description and legend in Fig.6). Figure 7 is still not informative nor explained in the text sufficiently. Delete it or rearrange accordingly. Figure 8 needs a proper legend to be meaningful.

Some of the Discussion contains Introduction information. Consider moving it there.

Materials and methods: The procedures were described in sufficient detail to allow repeating the experiments and analyses. I do not have major questions for this section.

Author Response

  1. Response to comment: 

There are numerous errors in the Abstract. This will not attract the attention of the potential readers to the article.

Response:

Thanks very much for the reviewer’s suggestion. We have made the corrections and supplements according to the reviewer’s suggestions. About the grammar and vocabulary errors raised by the reviewers, We have checked it carefully and corrected it in blue.

  1. Response to comment:

Results: Figure 3 is not visible even under 400% magnification. Consider changing the way of presenting these results. Improve the descriptions of Figures 4, 5 and 6 (preten-coding region, B-D panel description in Fig.5, visibility, A-D panel description, Y axis description and legend in Fig.6). Figure 7 is still not informative nor explained in the text sufficiently. Delete it or rearrange accordingly. Figure 8 needs a proper legend to be meaningful.

Response: 

Thanks very much for your suggestion. We have changed the format of Figure 3 and uploaded the original image as an attachment. The panel description of Figure 4-6 have been supplemented. Figure 7 mainly introduced the selection pressure on these four species during the evolution. The Ka/Ks of these 15 PCGs were all less than 1, indicating that PCGs were subject to purifying selection. The legand of Figure 8 have been supplemented.

  1. Response to comment:

Some of the Discussion contains Introduction information. Consider moving it there.

Materials and methods: The procedures were described in sufficient detail to allow repeating the experiments and analyses. I do not have major questions for this section.

Response: We have moved some information of Discussion Part into Introduction and the first paragraph of Discussion was deleted.

Reviewer 2 Report

This manuscript is a revision of a previously reviewed manuscript comparing the mitochondrial genomes of 4 Clonostachys fungal species. The manuscript is much improved and authors address multiple comments. My biggest comment is that the text needs to be reviewed again very closely for grammatical accuracy. I have listed a few instances below but there are more spread throughout the manuscript.

Line 34: Clonostachys can be reduced to C.

Line 48: Bionectriaceae Foundation?

Lines 54-55: Sentence requires revision.

Lines 65-74: This new text is not grammatically correct, for instance, the second sentence (Lines 66-67) makes no sense, and 4 mitogenomes were sequenced but both are reported. It should state that all 4 are reported.

Line 80: Need a space between "description" and [15.

Lines 78-82: New text is not grammatically correct and spacing of numbers and units is incorrect.

Figures 4 and 5 legends: "A" panes are labeled but B, C, and D panes are not. 

Author Response

  1. Response to comment:

This manuscript is a revision of a previously reviewed manuscript comparing the mitochondrial genomes of 4 Clonostachys fungal species. The manuscript is much improved and authors address multiple comments. My biggest comment is that the text needs to be reviewed again very closely for grammatical accuracy. I have listed a few instances below but there are more spread throughout the manuscript.

Line 34: Clonostachys can be reduced to C.

Line 48: Bionectriaceae Foundation?

Lines 54-55: Sentence requires revision.

Lines 65-74: This new text is not grammatically correct, for instance, the second sentence (Lines 66-67) makes no sense, and 4 mitogenomes were sequenced but both are reported. It should state that all 4 are reported.

Line 80: Need a space between "description" and [15.

Lines 78-82: New text is not grammatically correct and spacing of numbers and units is incorrect.

Figures 4 and 5 legends: "A" panes are labeled but B, C, and D panes are not.

Response:

Thanks very much for the reviewer’s suggestion. We have made the corrections and supplements according to your suggestions. Some irregularities in the paper have been corrected, such as grammar, abbreviations and format, We have checked it carefully and corrected it in blue. In addition, Bionectriaceae is a family of Hypocreales rather than Bionectriaceae Foundation. Figures 4 and 5 legends  have been supplemented. We have checked it carefully and corrected it in blue.

Your advice was a great help for us, thanks again for your advice.

This manuscript is a resubmission of an earlier submission. The following is a list of the peer review reports and author responses from that submission.

Round 1

Reviewer 1 Report

This manuscript compares mitochondrial genome sequences of 4 species of Clonostachys fungal species and details their phylogeny. The order Hypocreales is an ubiquitous and important fungal order that is lacking in details regarding species identity, function, and phylogeny. Thus, this work is important and interesting. However, the manuscript is sloppy and full of errors.

Abstract:

Page 1, Line 14: fungus should be fungi.

Line 19: Remove the word were

Line 21: Remove the word the.

Line 23: gene should be plural and remove the word were

Line 23: I do not understand "largest and minimum genetic distance".

Introduction:

Page 1, Line 35: C. araucaria should be italicized.

Line 36: sentence needs revised.

Page 2, Line 43: Which members were transferred from what genus, and where is the reference?

Line 45: Schroers is mentioned but reference is not provided here.

Line 45: Bionectria should be italicized.

Line 49: Bionectriaceae Foundation? And Bionectriaceae is mispelled.

Results:

Page 4: Line 99: remove "and".

Line 100: metagenomes should not be plural.

Line 100: extra period after sp.

Line 112: Both the four? This does not make sense.

Lines 113-114: Genes are 1-3 bp in size?

Line 135: Codon should not be capitalized.

Page 5: Figure 4: Which chart is associated with which Clonostachys species?

Page 6, Line 163: need a space between and and 1.76%.

Line 166: Extra space after found and "of" should be removed.

Figure 8 seems unnecessary. There are so few differences that thy can easily be highlighted in the text.

Page 9, Line 213: Phylogenetic should not be capitalized.

Line 223: remove "a".

Discussion:

Page 9, Line 230: was should be is.

Page 10, Line 245: identified is misspelled.

Materials and Methods:

Page 11, Lines 305-307: This sentence needs revision.

Line 307: fungal should be fungi.

Lines 310-311: Wang reference needs a publication year.

Line 317: space before "by".

Line 323: sequencing were conducted should be sequences were obtained.

Line 324: define Q20.

Lines 324-325: Remove parentheses from sentence.

Lines 328-329: Issues with spacing.

Line 333: Defines PCG here.

Line 333: Capitalize exonerate.

Page 12: Line 338: Provide company and location for Lasergene 7.1.

Line 350: using should be used.

Lines 350 and 355: -10 should be superscript.

Check format for references. Some are inconsistent.

Reviewer 2 Report

The paper describes a study on sequencing and analysis of the mitochondrial genomes of four Clonostachys species. Unfortunately, the manuscript is poorly prepared and contains a lot of errors and mistakes in terms of grammar, style and overall language quality. This often makes the sentence hard to understand and following the story is almost impossible. Major complaints and suggestions are given below.

There are numerous errors in the Abstract. This will not attract the attention of the potential readers to the article.

Introduction lacks focus. In fact, I was not sure what for was the study conducted, as it is no justification for the research undertaken, nor clear scientific problem to be solved. Simply, this is something like: we got four Clonostachys mitogenomes sequenced and compared and here they are. This is not enough for an article in a prestigious international journal. Again, language quality needs to be improved.

Results: there are no results for species identification shown nor cited. In Figure 2 only strain designations were used and no Accession numbers for sequences used in the analyses. This should be included, at least in the supplementary data. Figure 7 is not informative at all nor explained in the text sufficiently. Legend is also not satisfactory. Delete it or rearrange accordingly. Figure 8 is actually a Table and is not informative as well. Consider changing it anyhow to make it meaningful.

Discussion contains a lot of Introduction information. Consider moving it there.

Materials and methods: Only here there is the reference to the Accession Numbers of the strains tested (sub-section 4.6). It should also be mentioned before, in the Results, particularly.